# Whole-Genome Sequence and Comparative Analysis of *Trichoderma asperellum* ND-1 Reveal Its Unique Enzymatic System for Efficient Biomass Degradation

**Fengzhen Zheng [1], Tianshuo Han [2], Abdul Basit [3], Junquan Liu [2], Ting Miao [2] and Wei Jiang [2,*]**

[1] College of Biological and Environmental Engineering, Zhejiang Shuren University, 36 Zhoushan E Rd, Hangzhou 310015, China; 18811068358@163.com
[2] State Key Laboratory of Agro-Biotechnology, College of Biological Sciences, China Agricultural University, Yuan Ming Yuan West Road No. 2, Haidian, Beijing 100193, China; hants@cau.edu.cn (T.H.); ljq0802040421@163.com (J.L.); 15501017617@163.com (T.M.)
[3] Department of Microbiology, University of Jhang, Jhang 35200, Pakistan; abdul_9090@yahoo.com
[*] Correspondence: jiangwei01@cau.edu.cn; Tel./Fax: +86-10-62731440

**Abstract:** The lignocellulosic enzymes of *Trichoderma asperellum* have been intensely investigated toward efficient conversion of biomass into high-value chemicals/industrial products. However, lack of genome data is a remarkable hurdle for hydrolase systems studies. The secretory enzymes of newly isolated *T. asperellum* ND-1 during lignocellulose degradation are currently poorly known. Herein, a high-quality genomic sequence of ND-1, obtained by both Illumina HiSeq 2000 sequencing platforms and PacBio single-molecule real-time, has an assembly size of 35.75 Mb comprising 10,541 predicted genes. Secretome analysis showed that 895 proteins were detected, with 211 proteins associated with carbohydrate-active enzymes (CAZymes) responsible for biomass hydrolysis. Additionally, *T. asperellum* ND-1, *T. atroviride* IMI 206040, and *T. virens* Gv-298 shared 801 orthologues that were not identified in *T. reesei* QM6a, indicating that ND-1 may play critical roles in biological-control. In-depth analysis suggested that, compared with QM6a, the genome of ND-1 encoded a unique enzymatic system, especially hemicellulases and chitinases. Moreover, after comparative analysis of lignocellulase activities of ND-1 and other fungi, we found that ND-1 displayed higher hemicellulases (particularly xylanases) and comparable cellulases activities. Our analysis, combined with the whole-genome sequence information, offers a platform for designing advanced *T. asperellum* ND-1 strains for industrial utilizations, such as bioenergy production.

**Keywords:** *Trichoderma asperellum* ND-1; whole-genome sequencing; secretome; comparative genomics; lignocellulolytic enzymes; biomass degradation

## 1. Introduction

Lignocellulose from agricultural wastes, such as corn stover and sugarcane bagasse, serves as a widespread, renewable, and available resource [1–3]. Its components contain abundant and complex polysaccharides, including hemicellulose, cellulose, and lignin [4–6]. Particularly, hemicellulose and cellulose are becoming potential biomass feedstocks in the generation of high-value chemicals or bioenergy products [7–9]. Efficiently catalytic conversion of lignocellulose is mainly dependent on the availability of carbohydrate-active enzymes (CAZymes) [10,11], typically glycoside hydrolases (GHs), which degrade lignocellulosic biomass into simple sugars [12], a critical process for the production of second-generation bioethanol [13]. In spite of remarkable progress that has been achieved in enzymatic biodegradation of lignocellulosic materials [14,15], the high production cost of lignocellulases is still a major hurdle that must be solved prior to commercial-scale implementation of cellulosic ethanol [16].

In nature, the complete hydrolysis of biomass polysaccharides is usually carried out by synergetic action of various CAZymes (hemicellulases, cellulases, and lignin-modifying enzymes) rather than individuals [17–20]. Moreover, the discovery of lytic polysaccharide monooxygenases (LPMOs) has profoundly changed the way in which we view the enzymatic conversion of polysaccharides, particularly recalcitrant materials, such as cellulose and chitin [21]. LPMOs have been classified in the CAZymes database, within the Auxiliary Activity (AA) families AA9-11 and AA13-16, on account of their sequences [22–24]. The most widely investigated LPMO families are AA9 and AA10 [25]. LPMOs are currently known to be encoded in genomes across all kingdoms of life, especially fungi, and catalyze cleavage of various substrates [26–29]. In addition, LPMOs may be subjected to various post-translational modifications depending on their origin with effects on protein function and stability [30].

Due to their high specific enzyme activities and relatively strong protein secretion ability, several *Trichoderma* species and their cultivation on diverse agricultural wastes to yield polysaccharide-degrading enzymes have been intensely studied in the past two decades [31–33]. For example, *T. reesei* has been widely utilized in industrial fields and is regarded as a major source of commercial cellulases [34–36]. However, the enzymatic mixtures produced by *T. reesei* is deficient in high efficiency hemicellulases and other accessory enzymes, which facilitated investigations of other enzymes and/or fungi [37]. Among the numerous filamentous fungi that secreted hemicellulases and cellulases, *T. asperellum* is also known for its strong lignocellulosic hydrolysis ability [38–40]. Moreover, most of the investigations on *T. asperellum* were concentrated on the selection and expression of specific genes related to biomass-degrading enzymes, such as xylanases, β-glucanases, and cellobiohydrolases [11,38,41]. However, the genome sequence information of *T. asperellum* to further understand secretory proteins and carbohydrate-active enzymes remains to be fully investigated.

Rapid developments in sequencing strategies have facilitated the improvement of reference genomes in distinct microorganisms and the analysis of genome-scale variations [42]. The achievements have accelerated investigation for sequencing several *Trichoderma* genomes (including *T. reesei*, *T. virens*, and *T. atroviride*), utilizing genome shotgun approach, which provided a platform for detection of genome-wide differences and understanding degradation mechanisms of plant biomass polysaccharides [43,44]. Moreover, the PacBio RS sequencing platform has emerged to be the most advanced third-generation sequencer in the market in 2011 [45,46]. The system utilizes a unique and novel single molecule real-time (SMRT) detection technology that promotes the production of sequences with longer reads and decreases the level of bias [47–49]. Therefore, the application of PacBio sequencing technology provides a promising strategy to obtain advanced and accurate assemblies for *Trichoderma* genomes.

The goal of this study is to detect potentially significant enzymes and obtain an in-depth understanding of the lignocellulose-degrading mechanism from *T. Asperellum* ND-1. Therefore, the whole genome sequencing and comparative analysis of *T. Asperellum* ND-1 were performed by PacBio RS sequencing technology, with particular emphasis on biomass hydrolysis-related genes. Here, we described lignocellulolytic enzyme characteristics of *T. asperellum* ND-1 and compared them with other fungi (particularly *T. reesei*). Secretome analysis of *T. asperellum* ND-1 was also carried out to identified extracellular CAZymes. In addition, the diversity comparative analysis of the CAZymes in the genome of *T. asperellum* ND-1 and other fungi were achieved to provide novel insights into the biomass-decomposing enzymatic system of this fungus.

## 2. Results and Discussion

**Genome features of *T. asperellum* ND-1.** Whole-genome sequence of *T. asperellum* ND-1 was determined using PacBio and Illumina Hiseq × 10 platforms. De novo assembly using SOAPdenovo (version 2.04) and CANU (version 1.7) resulted in 32 scaffolds with N50 size of 2,032,888 bp (each with a length >1000 bp and N90 value of 792,243 bp). The

largest scaffold size was 3.7 Mb. The sum of the assembly length was 35.75 Mb, with a coverage of 99.3% (Table S1). Protein-coding genes, using the MAKER annotation method, yielded 10,541 genes for *T. asperellum* ND-1 and 12,802 genes for *T. asperellum* CBS 433.97, respectively, both greater than the estimate for *T. asperellum* IC-1 (8803) (Table S1).

The average gene density in the *T. asperellum* ND-1 genome was 290 genes per Mb. The average gene length was 1.86 (kb) and consisted of an average of 0.52 (kb) of the coding region and 0.16 (kb) of the non-coding region (Table 1), which were similar to other *Trichoderma* fungi [43]. The overall G+C content of the predicted genes was approximately 52.8%. tRNAScan-SE [50,51] identified a sum of 246 tRNAs containing 21 types of tRNAs in the genome (Table 1). Gene Ontology (GO) mapping was performed to detect GO terms for BLASTP functionally analyzed ORFs. BLASTP generated 6669 genes according to GO (Figure 1). Among them, 5092 genes belonged to molecular function (MF) (Figure 1A), 4325 genes were assigned to cellular component (CC) (Figure 1B), and 4903 genes were distributed into the biological process (BP) (Figure 1C). The major GO terms were constructed by the following groups: metabolic process (56.6%), catalytic activity (50.8%), cellular process (45.4%), binding (42.2%), cell (41.4%), cell part (41.2%), single-organism process (33.8%), organelle (33.7%), membrane (33.2%), and membrane part (30.5%). The genome size (35.75 Mb), total number of predicted genes (10, 541), and % (G+C) contents (48.65%) (Table 1) of *T. asperellum* ND-1 are comparable to the hypocreales mesophilic ascomycete fungus *T. reesei* (33.9 Mb) [44]. In addition, PFAM domains (7281) and proteases (82) of *T. asperellum* ND-1 genome were identified.

**Table 1.** Genome assembly and annotation statistics of *T. asperellum* ND-1.

| Featuers | *Trichoderma asperellum* **ND-1** |
|---|---|
| Coverage | 99.3% |
| Protein length, amino acids | 516.18 |
| Avg. Gene Density (genes/kb) | 0.29 |
| Avg. Gene length (bp) | 1.86 kb |
| Repeat Content % | 1.66 |
| tRNAs | 246 |
| Secreted Proteins | 895 |
| PHI genes | 2340 |
| Proteases | 82 |
| Average exons per gene | 2.98 |
| Average exon length (bp) | 0.52 kb |
| Average introns per gene | 1.98 |
| Average intron length (bp) | 0.16 kb |
| Supported by homology, Swissprot | 6746 (64%) |
| Supported by homology, NR | 9496 (90%) |
| Has PFAM domain | 7281 (69%) |

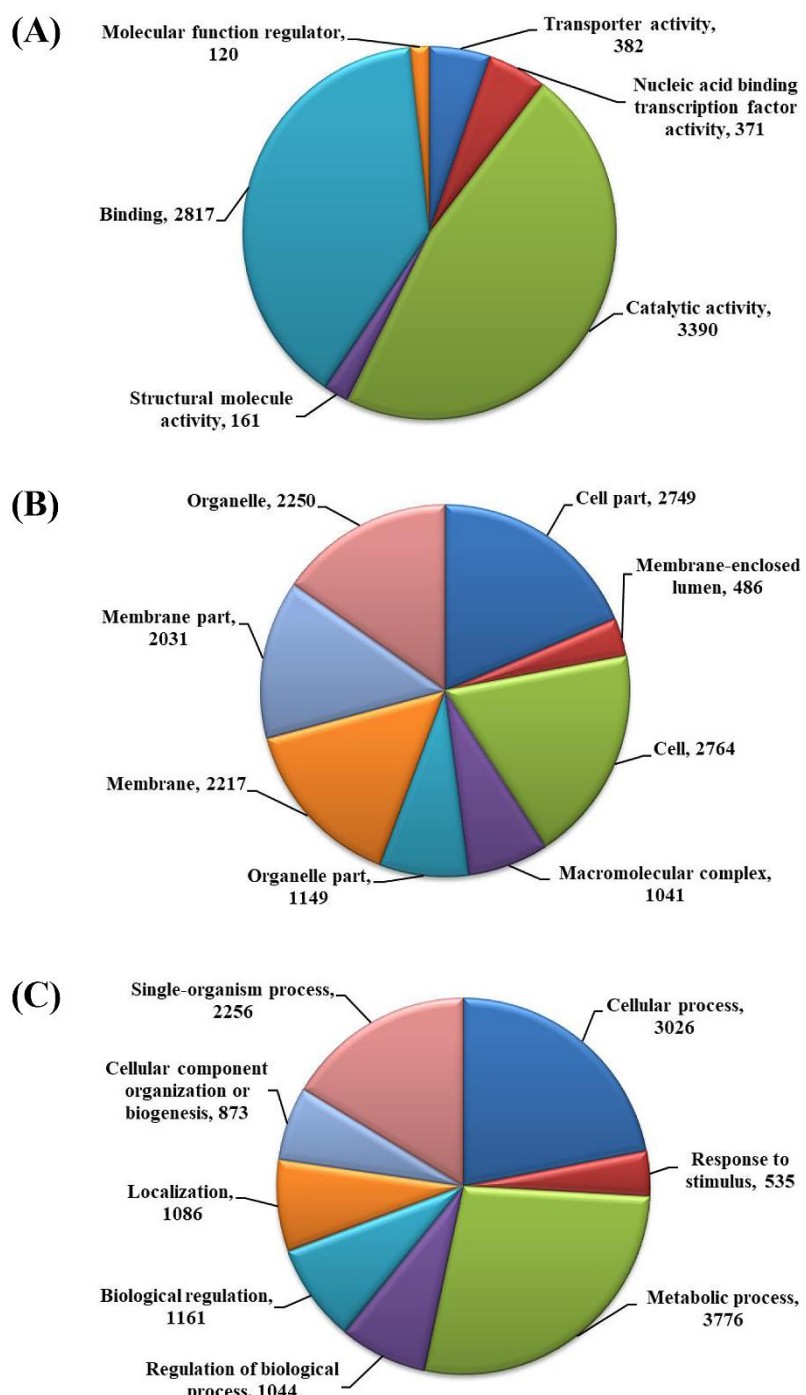

**Figure 1.** Gene ontology (GO)-based functional annotation of genes present in the *T. asperellum* ND-1 genome. (**A**) Molecular function domain (MF); (**B**) cellular process domains (CC); (**C**) biological process domains (BP).

**Mobile elements.** Repetitive DNA elements and transposable elements (TEs) play critical roles related to the gene functions, the evolution, and genome structure of the filamentous fungi [52]. The repeated sequences of the *T. asperellum* ND-1 genome were identified to be approximately 591,590 bp, including simple repeats, low complexity, small RNA, interspersed repeats, and satellites (Table S2). The repeated sequences represent 1.66% of the genome. Moreover, 78% of the TEs were simple repeats, whereas LINEs was just estimated to be 2% (Figure S1, Supplementary Materials). Notably, low complexity and small RNA account for 16 and 4%, respectively.

**Prediction and analysis of the *T. asperellum* ND-1 secretome.** The secretory proteome is believed to have an important role in identifying the capacity of the fungi to interact with distinct nature environment [38]. According to online software SignalP (version 4.1), the total number of 895 (represent 8.5% of the protein-coding genes) secreted proteins were predicted and annotated in *T. asperellum* ND-1 genome, which was higher than that of *T. reesei* Rut C30 (636 proteins) [53]. From this, GO terms were identified into 529 putative secreted proteins in the GO groups, namely, biological process (730), molecular function (561), and cellular component (675) (Figure S2). In the cellular component group, secretory proteins for membrane and membrane part, cell and cell part, organelle and organelle part, extracellular region, and macromolecular complex were highly abundant. Within the biological process, including the metabolic process, cellular process, localization, single-organism process, biogenesis or cellular component organization, biological regulation, and regulation of biological process, responses to the stimulus were highly represented. Under molecular function category, proteins related to binding, nucleic acid binding transcription factor activity, transporter activity, catalytic activity, electron carrier activity, and antioxidant activity were most abundant.

As for potential pathogenesis-related proteins of *T. asperellum* ND-1 secretome, 175 secreted proteins identified within the PHI database were assigned to various categories. Among them, 81 (41%) proteins were associated with reduced virulence, (76) 38% proteins were of unaffected pathogenicity, (18) 9% proteins were related to increased virulence (hypervirulence), and (18) 9% were related to loss of pathogenicity (Figure 2). Cytochrome P450 (CYP450) monooxygenase superfamily is involved in numerous metabolisms of the filamentous fungi, including secondary metabolites, lifestyle, and pathogenicity [54–56]. In *T. asperellum* ND-1, 163 CYP proteins were confirmed, of which 99 showed homologous counterparts in the PHI database.

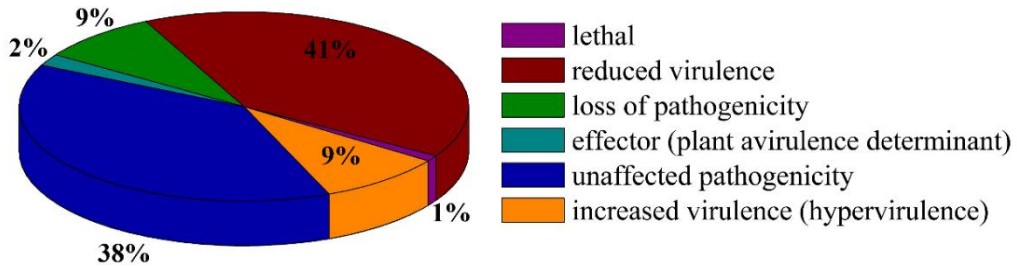

**Figure 2.** Summary of different phenotypic categories of orthologs of *T. asperellum* ND-1 secretome genes in the pathogen-host interactions (PHI-base) database.

A large number of extracellular enzymes secreted from *T. asperellum* have been recognized, many of which are involved in the degradation of complex biomass carbohydrates in various environments [38,41]. Using the CAZy database and carrying out a HMMER (version 3.3) scan, according to the profile compound in dbCAN release 2.0, we identified the presence of 67% GHs, 12% auxiliary activities (AAs), 10% carbohydrate esterases (CEs), 6% glycosyl transferases (GTs), 3% polysaccharide lyases (PLs), and 2% carbohydrate binding modules (CBMs) in *T. asperellum* ND-1 secretome (Figure 3A). LPMOs are the monocopper enzymes widely distributed in nature that catalyze the hydroxylation of glycosidic bonds in most abundant available polysaccharide in nature, i.e., cellulose [57,58]. Secretomic analysis revealed that *T. asperellum* ND-1 encode two predicted LPMOs from AA9 and AA11, respectively. Moreover, the AA9 family could have important roles as copper dependent LPMOs, cleaving oxidatively biomass cellulose [38]. Additionally, this work contributes to the broader mapping of enzyme activity in the Auxiliary Activity family (particularly AA9, AA11, and AA14) and provides new biocatalysts for potential applications in biomass modification.

The analysis of the CAZy categories was performed for the biomass hydrolysis enzymes families. Results showed that 141 genes encoding glycosyl hydrolases enzymes

were divided into 49 families. The GH families possessing three or more genes had 18 in *T. asperellum* ND-1 secretome, with GH18 being the largest family (17 genes), followed by GH16 (11 genes), GH55 (8 genes), GH3 (7 genes), GH92 (7 genes), and GH5 (6 genes) (Figure 3F). PL7 (3 genes), PL20 (2 genes), PL1 (1 genes), and PL8 (1 genes) of the PL families were also identified (Figure 3C). A previous study reported that members of the *Trichoderma* fungi (particularly *T. atroviride* and *T. harzianum*) are widely utilized as agricultural biocontrol agents [59,60], and both secondary metabolites and GH18 (chitinases) could play critical roles in growth and attacking pathogens [61]. Additionally, out of seven CEs families confirmed, members of family CE10 contained the maximum genes [7], followed by CE5 (4 genes), CE8 (3 genes), CE4 (2 genes), CE3 (2 genes), and CE1 (1 gene) (Figure 3E). The enzymatic activities of carboxylesterases were displayed in both CE10 and CE1 families [62]. Further, the enzymes providing auxiliary functions for degradation of polysaccharides were represented by four families of carbohydrate binding modules (CBM6, CBM24, CBM42, CBM66), glycosyl transferases (9 families) (Figure 3B), and auxiliary activities (9 families) (Figure 3D). Among them, the numbers of CBM42, AA7, and GT90/GT22/GT15 genes were significantly higher. Moreover, the secretory proteins of *T. asperellum* ND-1 also contained an assortment of proteases, transferases, and chitinases. These results imply that *T. asperellum* ND-1 secretome consists of various functional proteins and the major components associated with proteolytic and cellulolytic enzymes, which are crucial for promoting the hydrolysis of the host plant to obtain essential nutrients and adapt various environments.

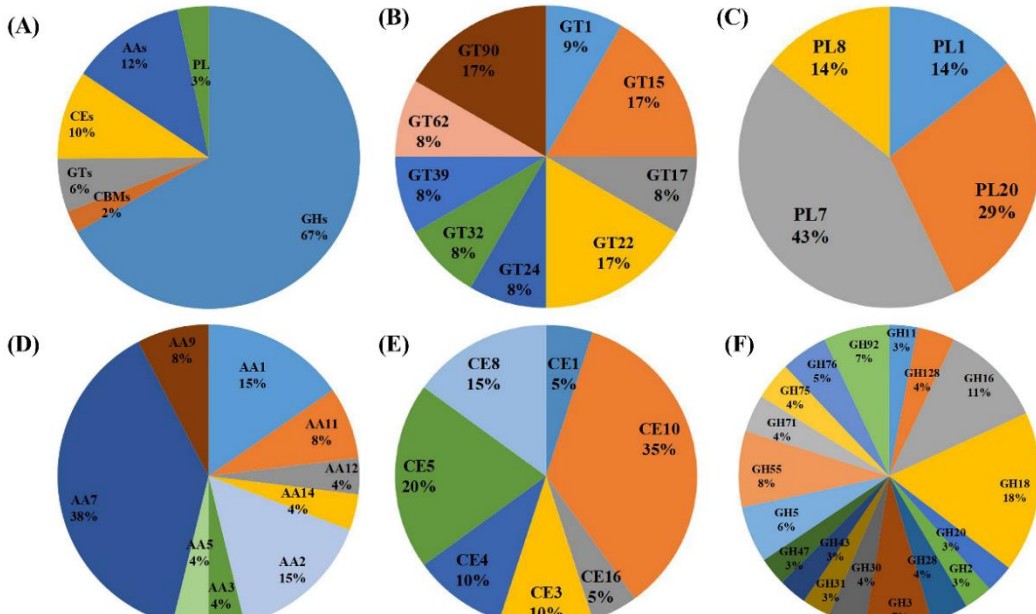

**Figure 3.** CAZymes identified in the secretome of *T. asperellum* ND-1. (**A**) Summary of the six CAZyme categories: auxiliary activities (AAs), carbohydrate- binding modules (CBMs), polysaccharide lyases (PLs), glycoside hydrolases (GHs), glycosyl transferases (GTs), and carbohydrate esterases (CEs). (**B**) Distinct summaries of the CAZyme GTs. (**C**) Distinct summaries of the CAZyme PLs. (**D**) Distinct summaries of the CAZyme AAs. (**E**) Distinct summaries of the CAZyme CEs. (**F**) Distinct summaries of the CAZyme GHs.

**Phylogenetic relationships.** The evolutionary relationships of *T. asperellum* ND-1 and other selected fungi species were evaluated using the proteomes of these fungi. According to phylogenetic analysis results, all the selected *Trichoderma* were distributed into a single primary cluster (Figure 4). Majority of *Trichoderma* species are commonly applied in agriculture as effective agents for biological control against many phytopathogenic microorganisms; examples are *T. asperellum* T203 [63], *T. harzianum* [59], and *T. asperellum*

SKT-1 [64]. The isolated lignocellulolytic fungus *T. asperellum* ND-1 is evolutionarily close to *T. asperellum* CBS 433.97 (Figure 4). *T. asperellum* ND-1 is also close to the other two biological species, *T. atroviride* IMI 206040 and *T. gamsii* T6085, suggesting that *T. asperellum* ND-1 may have biocontrol functions applied in agriculture. Moreover, *T. asperellum* ND-1 and *T. reesei* QM6a (a representational producer of plant biomass degrading enzymes) were distributed into different subclusters (Figure 4). In addition, all the *Asperellus* species were grouped into another single clade that is distantly related to *T. asperellum* ND-1. *Grifola frondosa* 9006-11 was served as an outgroup in the phylogenomic analysis.

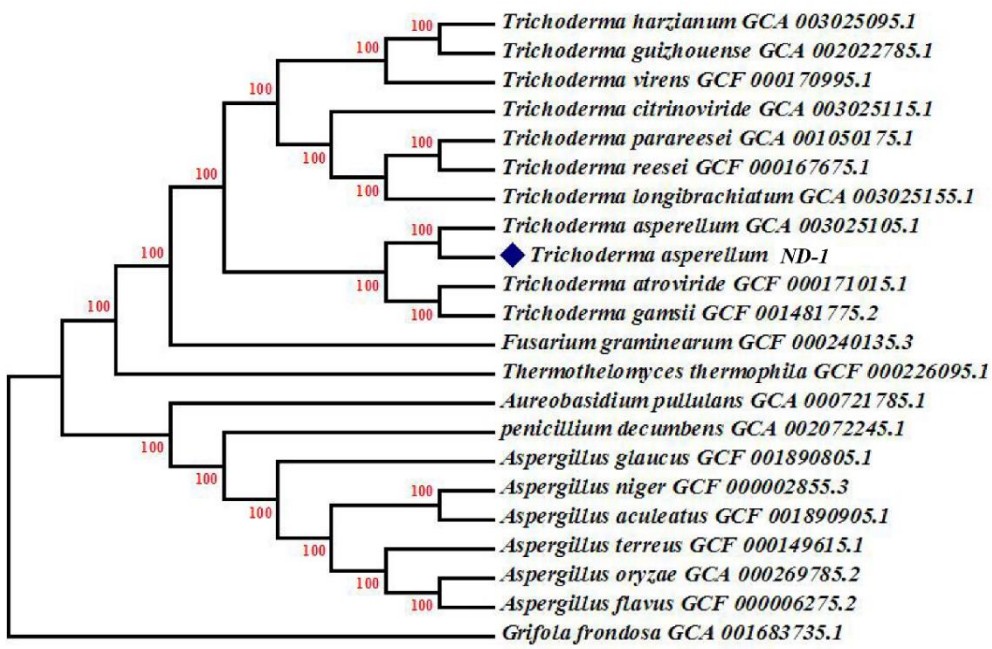

**Figure 4.** Whole-genome phylogenetic analysis of *T. asperellum* ND-1 sequences. The tree of selected genome sequences was constructed by using the neighbor-joining (NJ) method with the Poisson model, as implemented in Mega software 7.0. *T. asperellum* ND-1 was labeled with diamonds.

**Comparative analysis of orthologous genes between different *Trichoderma* species.** The annotated proteome of *T. asperellum* ND-1 was further compared with the other three biological species, *T. reesei* QM6a, *T. virens* Gv-298, and *T. atroviride* IMI 206040, by using orthoMCL [65]. Among the four *Trichoderma* species, a total of 7073 common clusters were identified (Figure 5). The common clusters accounted for 60–80% of the four fungal proteomes, respectively, which indicated that the vast majority of the genes were conserved in the *Trichoderma* group. However, *T. asperellum* ND-1, *T. atroviride* IMI 206040, and *T. virens* Gv-298 contained about 1381, 1618, and 1991 species-specific clusters, respectively, but the *T. reesei* QM6a had only 520 unique clusters (Figure 5), consistent with a previous study, showing that *T. reesei* contained fewer exclusive orthologous genes than other sequenced fungus [43,44]. Moreover, *T. virens* and *T. atroviride* are probably the most popular investigated biocontrol agents utilized in various agriculture fields [60]. In this study, we found that *T. asperellum* ND-1, *T. atroviride* IMI 206040, and *T. virens* Gv-298 shared 801 orthologues that were not detected in *T. reesei* QM6a (Figure 5), which may be partial factors that resulted in a *T. asperellum* ND-1 biological control function [63,64]. In addition, we identified that 7250 orthologous genes were present between *T. asperellum* ND-1 and *T. reesei* QM6a (Figure 5), indicating that *T. asperellum* ND-1 may have strong biomass degradation ability [38,41]. A number of 7889, 8179, and 8957 common clusters were also predicted between *Trichoderma* species (Figure 5) when comparing *T. reesei* QM6a vs. *T. atroviride* IMI 206040, *T. reesei* QM6a vs. *T. virens* Gv-298, and *T. atroviride* IMI 206040 vs. *T. virens* Gv-298, respectively.

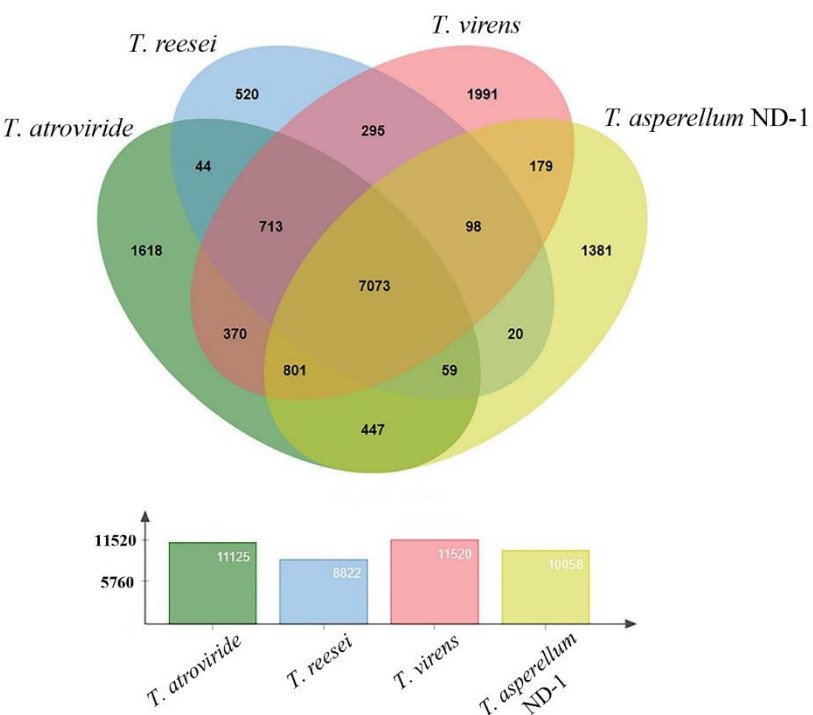

**Figure 5.** Distribution of orthologues of *T. atroviride* IMI 206040, *T. virens* Gv-298, *T. reesei* QM6a, and *T. asperellum* ND-1.

**Diversity of carbohydrate-active enzymes in *T. asperellum* ND-1 and other fungi.** The components of lignocellulosic biomass contain structural polysaccharides, e.g., cellulose, xylan, and mannan [6,13,16]. CAZymes that hydrolyzed the poly- and oligosaccharides play a critical role in the biology of filamentous fungi [38,40]. The *T. asperellum* ND-1 genome harbored 438 genes encoding for CAZymes, the members of which were identified with the presence of 83 candidate GTs from 29 families, 59 AAs from 11 families, 41 CEs from 8 families, and 14 CBMs from 9 families, in addition to 7 PLs from 4 families (Figure S3). The largest group of CAZymes were GHs (234 genes), which were categorized into 57 various families. The size of the GHs in *T. asperellum* ND-1 was close to *T. atroviride* IMI 206040 (242 genes) and *T. virens* Gv-298 (250 genes) (Table S3).

To further evaluated biomass-degrading abilities of *T. asperellum* ND-1, the diversity of CAZymes in *T. asperellum* ND-1 was compared with other fungi (particularly *T. reesei and T. asperellum* CBS 433.97) (Figure 6). The number of CAZyme-encoding genes and GH class distribution among the five *Trichoderma* species were remarkably different (Figure 6). *T. reesei* QM6a, a well-known biomass polysaccharide degrader, possesses a variety of genes encoding GHs [44]. However, with a total of 189 GH encoding genes, it has fewer GHs than the *T. asperellum* ND-1 (234 genes) (Table S3). Analysis of the *T. asperellum* ND-1 for CAZymes predicted various proteins involved in xylan degradation. For example, three endo-1,4(3)-β-xylanases (representing families GH10) and four endo-1,4(3)-β-xylanases (GH11) were identified for *T. asperellum* ND-1, as opposed to one endo-1,4(3)-β-xylanase (GH10) and three endo-1,4(3)-β-xylanases (GH11) in *T. reesei* QM6a (Figure 6). *T. asperellum* CBS 433.97 also contained two endo-1,4(3)-β-xylanases (representing families GH10) and four endo-1,4(3)-β-xylanases (GH11) (Figure 6). Xylanase adding to cellulase mixtures has significant improvement for complete degradation of lignocellulosic biomass, on account of enhancing the cellulase's accessibility to cellulose [17,19,66]. However, the component of xylan in lignocellulosic polymers has a backbone of xylose units, which can be connected with various residues [20]. Complete degradation of heteroxylan needs a battery of side-chain-degrading enzymes [67], including GH2 (β-glucuronidase or β-mannanase), GH5 (endo-β-1,4-mannase), GH27 (α-galactosidase), GH28 (α-L-

rhamnosidase), GH62 ($\alpha$-L-arabinofuranosidase), GH67 ($\alpha$-glucuronidase), GH76 ($\alpha$-1,6-mannanase or $\alpha$-glucosidase), GH78 ($\alpha$-L-rhamnosidase), GH92 ($\alpha$-1,2(3)-mannosidase), GH125 (exo-$\alpha$-1,6-mannosidase), and GH154 ($\beta$-glucuronidase), which were predicted in all five *Trichoderma* species (Figure 6). In addition, GH1 was mainly composed of $\beta$-glucosidase and $\beta$-galactosidase, and the two GH1 enzymes of *T. reesei* QM9414 were $\beta$-glucosidase [68], which was similar with the present study. In *T. asperellum* ND-1 and *T. asperellum* CBS 433.97, $\alpha$-mannosidases, $\alpha$-L-arabinofuranosidases, and $\beta$-xylosidases were identified in higher abundances (Figure 6). For example, the genome of *T. asperellum* ND-1 and *T. asperellum* CBS 433.97 contained eight genes encoding GH31 enzymes with $\alpha$-mannosidase or $\alpha$-xylosidase activities (Figure 6). A number of GH43 ($\beta$-xylosidase or $\alpha$-L-arabinofuranosidase) were also remarkably expanded in *T. asperellum* CBS 433.97, *T. asperellum* ND-1, and *T. atroviride* IMI 206040. Enzymes from other GH families, such as GH32 (arabinosidase), GH93 (exo-$\alpha$-L-1,5-arabinanase), and GH114 (endo-$\alpha$-1,4-polygalactosaminidase), were detected in the genome of *T. asperellum* CBS 433.97, *T. atroviride* IMI 206040, *T. virens* Gv-298, and *T. asperellum* ND-1 (alongside *T. reesei* QM6a) (Figure 6). The diversity of hemicellulases in *T. asperellum* ND-1 was much larger in some respects than other biomass-degrading fungi (especially *T. reesei*), which were similar to previous studies [38,44]. Moreover, hydrolases from GH51 ($\alpha$-L-arabinofuranosidase or $\beta$-xylosidase), GH127 ($\beta$-L-arabinofuranosidase), and GH142 ($\beta$-L-arabinofuranosidase) were predicted in the *T. asperellum* CBS 433.97, *T. asperellum* ND-1, and *T. atroviride* IMI 206040 genome only (Figure 6). In addition, the genome of *T. asperellum* CBS 433.97 contained nine genes encoding GH27 enzymes with $\alpha$-galactosidase activities, which was three-fold higher than that of *T. asperellum* ND-1 (Figure 6). These findings, taken together, revealed that *T. asperellum* ND-1 generated more diversity of CAZymes relevant to hemicellulose hydrolysis than those of *T. reesei* QM6a.

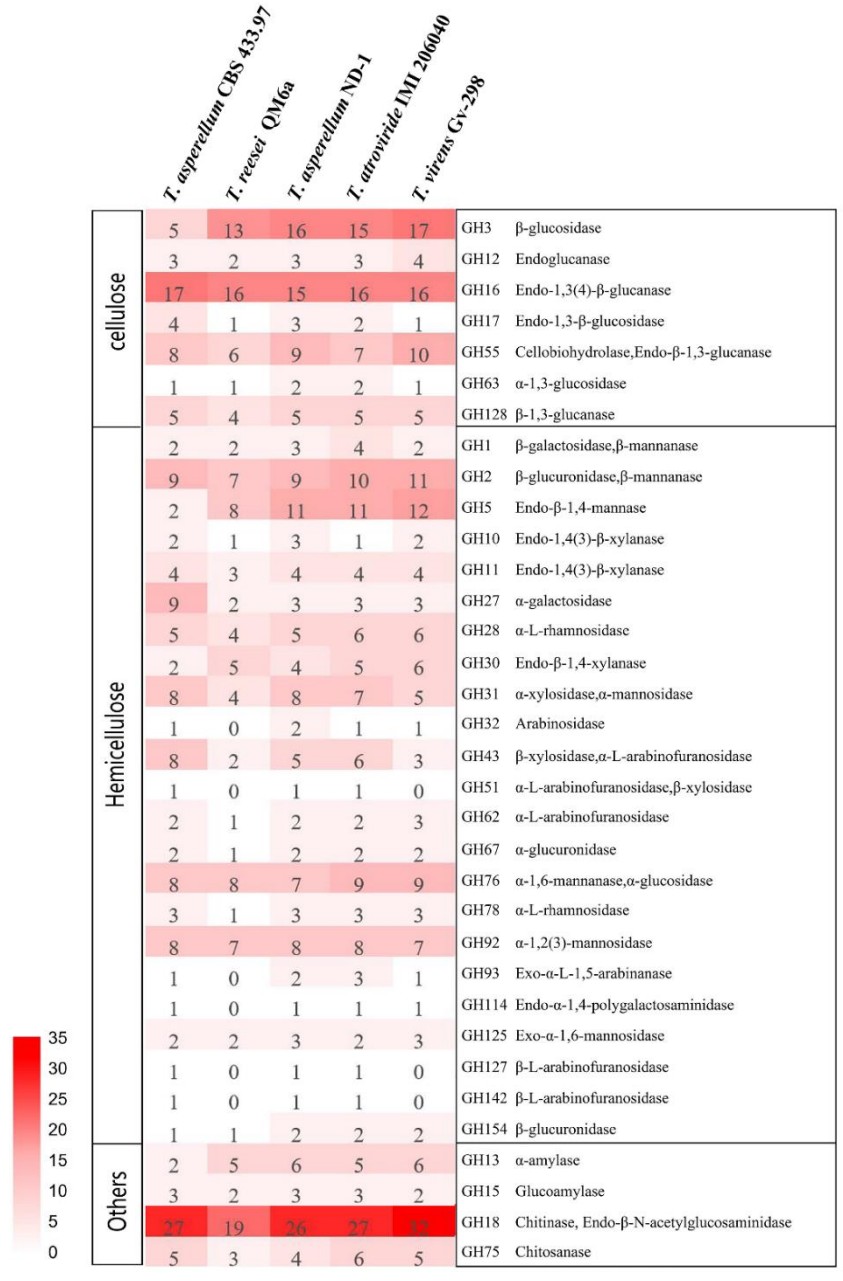

**Figure 6.** Heat map showing the distribution of glycoside hydrolases (GHs) in the genomes of *T. atroviride* IMI 206040, *T. virens* Gv-298, *T. reesei* QM6a, *T. asperellum* ND-1, and *T. asperellum* CBS 433.97. CAZymes families are grouped according to their activities in major components of plant cell walls.

Degradation of complex cellulose polysaccharides depends on synergistic actions of three representative cellulases: β(α)-glucosidases, exo-1,4-β-glucanases, and endo-1,4-β-glucanases [69,70]. A much higher number of enzymes associated with cellulose degradation were identified in the genome of *T. virens* Gv-298, *T. asperellum* ND-1, and *T. atroviride* IMI 206040 (54, 53, and 50 proteins, respectively), compared with that of *T. reesei* QM6a and *T. asperellum* CBS 433.97 (Figure 6). Three endoglucanases (GH12), five β(α)-glucosidases (GH17 and GH63), and three exo-1,4-β-glucanases (GH6/7) were predicted in the *T. asperellum* ND-1 genome, and three endoglucanases (GH12), four β(α)-glucosidases (GH17 and GH63), and three exo-1,4-β-glucanases (GH6/7) were identified in the genome of *T. atroviride* IMI 206040 (Figure 6). Particularly in *T. virens* Gv-298, 17 β-glucosidases (GH3), four endoglucanases (GH12), and three exo-1,4-β-glucanases (GH6/7) were the most abundant hydrolases (Figure 6), compared to the other described fungi. Moreover,

17 endo-1,3(4)-β-glucanase (GH16) and four endo-1,3-β($\alpha$)-glucosidase (GH17) were identified in the genome of *T. asperellum* CBS 433.97. In contrast, detected cellulolytic enzymes of *T. reesei* QM6a contained only two endoglucanases (GH12), two β($\alpha$)-glucosidases (GH17 and GH63), and three exo-1,4-β-glucanases (GH6/7) (Figure 6), consistent with a previous study, which showed that the genome of *T. reesei* contained fewer cellulase-related genes than other *Trichderma* fungi [38,43,44]. In addition, the number of CAZyme-encoding genes and GH class distribution among the different *Trichoderma* species were remarkably different. *T. reesei* Rut-C30 had at least two exo-1,4-β-glucanases (Cel6a: GH6 and Cel7a: GH7), four endo-1,4-β- glucanases (Cel5a: GH5, Cel7b: GH7, Cel12b: GH12, and Cel45a: GH45), and one β-glucosidase (Cel3a: GH3) [71]. In recent years, the characteristics of 11 β-glucosidases (two GH1 enzymes and nine GH3 enzymes) has been analyzed [68,72]. Moreover, a high number of GH3, GH16, and GH128 enzymes suggested that *T. asperellum* ND-1 has a larger substrate range, which can be applied for various applications, including biomass conversion and biofuel production.

All five *Trichoderma* species also produced a large series of enzymes, the majority of which were known to be associated with chitin degradation. For example, the GH18 family, containing various enzymes linked to chitin hydrolysis [61], was remarkably expanded in the genomes of *T. virens* Gv-298, *T. atroviride* IMI 206040, *T. asperellum* CBS 433.97, and *T. asperellum* ND-1 (32, 27, 27, and 26 genes, respectively), relative to *T. reesei* QM6a (19 genes) (Figure 6). The component of fungal cell walls was comprised of substantial chitin and chitinolytic enzymes and was therefore an indispensable part of mycoparasitic attack [48]. Moreover, hydrolases from GH75 (chitosanases) and GH18 (endo-β-N-acetylglucosaminidases) also play a critical role in the degradation of fungal cell walls [43,61]. The most abundant of all glycoside hydrolases in *T. asperellum* ND-1 genome was GH18 comprised of 26 chitinolytic enzymes (Figure 6), which is consistent with a previous study [73]. Therefore, the *T. asperellum* ND-1 may be served as an effective and environmentally friendly bio-control agent, similar to *T. virens* Gv-298 and *T. atroviride* IMI 206040, against numerous phytopathogenic microorganisms [43]. In addition, identified amylolytic enzymes of *T. asperellum* ND-1 comprised six $\alpha$-amylase (GH13), and three glucoamylase (GH15) were detected (Figure 6). Consequently, *T. asperellum* ND-1 could have great application potentials in the production of value-added biomolecules maltose from $\alpha$-glucan like starch.

**Comparative analysis of lignocellulolytic enzyme activities.** Efficiently catalytic degradation of lignocellulose is dependent on the synergistic action of various enzymes that hydrolyze lignocellulolytic biomass into fermentable sugars [17,19,20]. The present results show that *T. asperellum* ND-1 and other filamentous fungi displayed different time course profiles of lignocellulase activities (Figure 7).

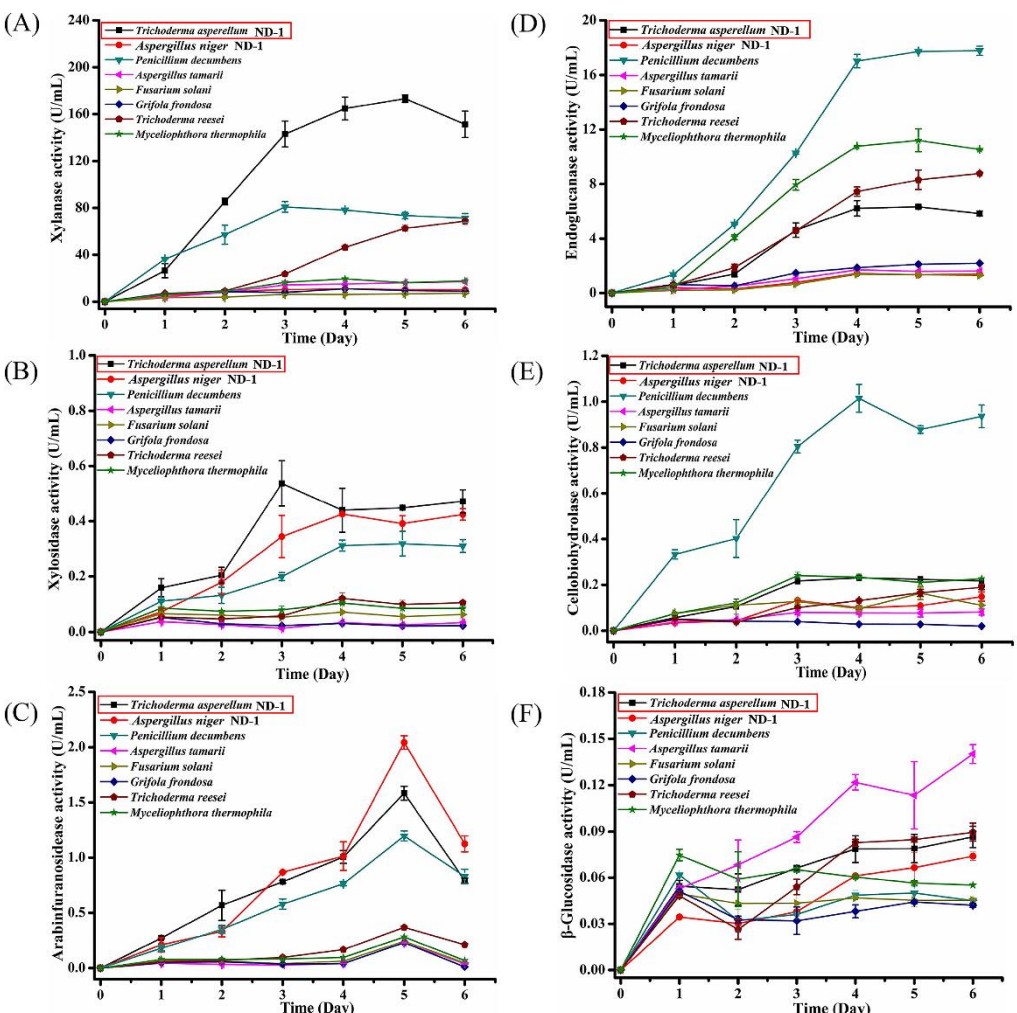

**Figure 7.** Comparison of lignocellulolytic enzyme activities produced by *T. asperellum* ND-1 and other fungi. The activities of hemicellulases (endoxylanase, β-xylosidase, and α-L- arabinofuranosidase) are shown in (**A–C**), respectively. The activities of cellulases (endoglucanase, cellobiohydrolase, and β-glucosidase) are shown in (**D–F**), respectively.

Among hemicellulases, xylanase activity in *T. asperellum* ND-1 extract improved sharply over time and obtained the highest level of 173.25 ± 3.14 U/mL after 5 days of cultivation (Figure 7A). Xylanases produced by *T. asperellum* ND-1 were identified to efficiently hydrolyze xylan into major product xylobiose [11]. For *P. decumbens*, the corresponding activity increased slowly to reach the maximum value (80.83 ± 4.55 U/mL) on day 3 (Figure 7A). Xylanase activity produced by *T. reesei* increased gradually until the end of the cultivation and a maximum of 68.84 ± 2.98 U/mL was achieved on day 6, while it was much lower in *G. frondosa*, *F. solani*, *A. tamarii*, *A. niger* ND-1, and *M. thermophila* extract (Figure 7A). β-xylosidase activity of *T. asperellum* ND-1 increased over time, and a peak (0.54 ± 0.08 U/mL) displayed on day 3 (Figure 7B). For *A. niger* ND-1 and *P. decumbens*, the activity was up to a maximum (0.43 ± 0.003 U/mL and 0.31 ± 0.02 U/mL, respectively) after 4 days (Figure 7B) and then remained relative stable. In contrast, the enzyme activity of *T. reesei*, *M. thermophila*, and *F. solani* fluctuated at a low level between 0.046 ± 0.003 and 0.12 ± 0.02 U/mL during the cultivation time (Figure 7B). A minimal β-xylosidase activity was observed in the *G. frondosa* and *A. tamarii* extract (Figure 7B). In addition, reports reveal that side-chain-degrading enzymes play a crucial role in the degradation of biomass [17]. α-L-arabinofuranosidase activities were found and showed the maximum level on day 5 in all selected fungi. In *T. asperellum* ND-1, the enzyme activity reached a maximum value of 1.58 ± 0.06 U/mL (Figure 7C), with a 5.5-fold, 4-fold, and 1.3-fold higher

level, respectively, compared with that of *M. thermophila*, *T. reesei*, and *P. decumbens*. These results, taken together, reveal that *T. asperellum* ND-1 produced various hemicellulases with significantly higher activities, which can be utilized in various fields, particularly in the production of valuable biomolecules (prebiotics, xylooligosaccharides).

In terms of cellulolytic enzymes, endoglucanases from *T. asperellum* ND-1 displayed the highest level (6.33 ± 0.11 U/mL) on day 5, and the activity exhibited was comparable with that of *T. reesei* (8.78 ± 0.03 U/mL) (Figure 7D). The corresponding enzyme activity in *P. decumbens* and *M. thermophila* increased rapidly to the maximum value (17.71 ± 0.03 U/mL and 11.20 ± 0.84 U/mL, respectively) on day 5 (Figure 7D), but it was too low in *A. niger* ND-1, *A. tamarii*, *F. solani*, and *G. frondosa* during the overall cultivation period. In addition, another two major cellulase activities (cellobiohydrolase (0.23 ± 0.01 U/mL) and β-glucosidase (0.087 ± 0.01 U/mL)), produced by *T. asperellum* ND-1, were much higher than those of other selected fungi (Figure 7E,F), including *A. niger* ND-1, *F. solani*, and *G. frondosa*. The maximum exoglucanase activity (1.01 ± 0.06 U/mL) in *P. decumbens* extract appeared on day 4 (Figure 7E), and the β-glucosidase secreted by *A. tamarii* obtained a maximum activity of 0.14 ± 0.01 U/mL on day 6 (Figure 7F). *T. reesei* was well known for its involvement in the degradation of complex biomass carbohydrates and was used as the main industrial producers of cellulases [34,35]. These lignocellulase activities profiles indicated that *T. asperellum* ND-1 generated an enzyme mixture with enhanced cellulose hydrolysis capability similar to that of *T. reesei*. Moreover, genome sequencing and analysis of the biomass-degrading fungus *T. asperellum* ND-1 were performed to pave the way for designing enhanced *T. asperellum* ND-1 strains toward a more rapid conversion of lignocellulose into soluble sugars for bioenergy production.

## 3. Conclusions

The whole genome sequence and lignocellulases activities of the newly isolated *T. asperellum* ND-1 were determined for the first time. A high-quality genomic sequence of ND-1 has an assembly size of 35.75 Mb comprising 10,541 predicted genes. Secretome analysis showed that 895 proteins were detected, with 211 proteins associated with CA-Zymes, possessing remarkable potential for utilization in biomass decomposition. Comparative genome analysis suggested that the genome of ND-1 contained many genes involved in biological-control, which would be useful to investigate *Trichoderma* species as biocontrol agents. Furthermore, the genome of ND-1 encoded a higher diversity of polysaccharide-degrading enzymes, especially those associated with hemicellulose deconstruction. Compared with *T. reesei* (CICC 40932), ND-1 produced higher hemicellulases (particularly xylanase) and similar cellulases activities. These results will help us understand the unique hydrolytic enzyme system of *T. asperellum* ND-1 and promote the investigation of more efficient and cost-effective enzymes for the degradation of lignocellulosic biomass.

## 4. Materials and Methods

**Strains, reagents, and media.** The *T. asperellum* ND-1 (GenBank accession number MH496612) and *A. niger* ND-1 (GenBank Accession number MH137707) strains were isolated from soil samples collected in Chifeng, Inner Mongolia, China, and preserved in the laboratory. The *T. reesei* (CICC 40932), *Grifola frondosa* (CICC 14078), *Penicillium decumbens* (CICC 40674), *Fusarium solani* (CICC 2618), and *A. tamarii* (CICC 40233) were obtained from the China Center of Industrial Culture Collection. *Myceliophthora thermophila* ATCC 42464 was from the American Type Culture Collection. P-nitrophenyl (pNP)-D-β-glucopyranoside (pNPG), pNP-L-α-arabinofuranoside (pNPAf), pNP-β-D-xylopyranoside (pNPX), pNP-D-β-cellobiose (pNPC), and sodium carboxymethyl cellulose (CMC-Na) were from Sigma-Aldrich (St. Louis, MO, USA). Beechwood xylan (BWX) was purchased from Megazyme (Wicklow, Ireland).

All fungi were precultured on potato dextrose agar (PDA) at 28 °C for 4 days. In total, 5 g of unpretreated, dry corn stover (milling to 2 cm), 0.2 g tryptone, 0.2 g yeast extract,

and 100 mL of a basal salt solution (0.5 g/L MgSO$_4$·7H$_2$O, 1 g/L K$_2$HPO$_4$·3H$_2$O, 2 g/L NH$_4$Cl, 0.5 g/L KCl, 0.02 g/L FeSO$_4$·7H$_2$O, 0.03 g/L CaCl$_2$, and 0.02 g/L ZnSO$_4$·7H$_2$O) were added to 250 mL Erlenmeyer flasks, and the mixtures were sterilized at 121 °C for 30 min used as inducing medium. For lignocellulases (cellulases, hemicellulases) activity analysis, suspensions of *T. asperellum* ND-1 and other fungi were inoculated onto sterile inducing medium at approximately 5 × 10$^8$ spores, which were cultured with agitation at 200 rpm, 28 °C, for 6 days. Culture samples were taken every day and centrifuged at 12,000× *g* for 10 min to collect the supernatant. The supernatant containing the crude enzymes was then used directly for enzyme assays. Experiments were performed in triplicate.

**Genomic DNA preparation and quality assessment.** The *T. asperellum* ND-1 (MH496612) strain cultured on PDA medium at 28 °C for 2 days was inoculated in potato dextrose broth (PDB) and incubated at 28 °C for 3 days, 200 rpm. Fungal biomass (3.5 g) of 500 mL was acquired via centrifugation for 15 min, at 4000 rpm, and maintained in liquid nitrogen. Genomic DNA of *T. asperellum* ND-1 was isolated using the Omega Fungal DNA Kit D3390-02, according to fungal DNA extraction protocol. The purity and concentration of genomic DNA were quantified the with NanoDrop 2000 (Thermo Fisher Scientific, Waltham, MA, USA) and TBS-380 (Turner BioSystems Inc., Sunnyvale, CA, USA) methods, respectively.

**Sequencing and assembly.** *T. asperellum* ND-1 genome was sequenced using a combination of PacBio sequel single molecule real-time (SMRT) [42] and Illumina sequencing platforms (MajorBio Co., Shanghai, China). DNA libraries containing ~400 bp and 10-kb inserts were prepared. The 400-bp library was constructed according to NEXTflex™ Rapid DNA-Seq Kit, including fragmentation of genomic DNA, end repair, adaptor ligation, and PCR amplification. The 400-bp library was used for paired-end Illumina sequencing (2 × 150 bp) by Illumina HiSeq 2000 and assembled with SOAPdenovo version 2.04 (http://soap.genomics.org.cn/, accessed on 5 April 2022). The 10-kb library was prepared using PacBio's standard methods. DNA fragments were purified, end-repaired, and ligated with SMRTbell sequencing adapters following the manufacturer's instruction (Pacific Biosciences, Menlo Park, CA, USA). The 10-kb library was evaluated with 2100 Bioanalyzer (Agilent, Santa Clara, CA, USA), sequenced by SMRT, and the sequencing results (filtered reads: 4.92 G, sequencing depth: 123×) were assembled into contigs through CANU (version 1.7) with default parameters [74]. Furthermore, error correction of the PacBio assembly results was performed using the Illumina reads and gap filling with GAPCLOSER version 1.12 [75]. Finally, quality assessment of genome assembly was carried out using CEGMA (version 2.5) and BUSCO (version 3.0) softwares.

**Gene prediction and annotation.** Genome prediction of protein-encoding sequences (opening reading frames, ORFs) were carried out by a combination of four independent softwares, GeneMark-ES (version 2.3a) [76], SNAP [77], MAKER (version 2.31.9) (http://www.yandell-lab.org/software/maker.html, accessed on 5 April 2022), and Augustus (version 2.5.5) (http://augustus.gobics.de/, accessed on 5 April 2022). The tRNAscan-SE version 2.0 was used for tRNA prediction (50,51). Gene annotations for predicted ORFs were carried out by various databases, including the Non-Redundant Protein database (NR) (ftp://ftp.ncbi.nlm.nih.gov/blast/db/, accessed on 5 April 2022), Swiss-Prot database (https://web.expasy.org/docs/swiss-prot_guideline.html, accessed on 5 April 2022) [78], COG (http://www. ncbi. nlm.nih.gov/COG/, accessed on 5 April 2022) [79], and KEGG database (http://www.genome. jp/kegg/, accessed on 5 April 2022) [80] using blastP with E-values of ≤ l × 10$^{-5}$. Proteins coding for proteases were classified by conducting Blastp (batch) against the MEROPS database (http://merops.sanger.ac.uk, accessed on 5 April 2022). Domain identification of predicted protein-encoding sequences were analyzed according to the Pfam database (http://pfam.xfam.org/, accessed on 5 April 2022) [81] and HMMER version 3.3 (http://www.hmmer.org/, accessed on 5 April 2022) [82]. The enrichment analysis of gene ontology (GO) was obtained by using Blast2GO version 2.5 (https://www.blast2go.com/, accessed on 5 April 2022) [83].

**Identification of transposable elements.** Transposable elements (TEs) containing various classes (LTRs (long terminal repeats), LINEs (long interspersed nuclear elements), DNA transposons, etc.) were determined strictly using three methods. We firstly detected the *T. asperellum* ND-1 with a de novo software Repeat Modeler (http://repeat-masker.org/RepeatModeler/, accessed on 5 April 2022), then the predicted repetitive elements were identified by BLASTP searches against the Repeat Protein Masker (www.repeatmasker.org/cgi-bin/RepeatProteinMaskRequest, accessed on 5 April 2022) and Repeat Masker (version 4.0.7) database-based softwares (http://www.repeatmasker.org/, accessed on 5 April 2022). All the parameters were set as default.

**Secretome prediction and analysis.** The identification of putative secreted proteins was carried out by SignalP version 4.1 (http://www.cbs.dtu.dk/services/SignalP, accessed on 5 April 2022). The Blast2GO version 2.5 (https://www. blast2go.com/, accessed on 5 April 2022) [83] and BLASTP analysis with E-values of $\leq 1 \times 10^{-5}$ were used for the functional annotations of predicted secretome, according to the terms "cellular component", "biological process", and "molecular function" in the GO database. Potential pathogenicity-related genes (related to reduced virulence, unaffected pathogenicity, lethal, loss of pathogenicity, etc.) were analyzed by detecting against the pathogen-host interaction (PHI) database (http://www.phi-base.org/, accessed on 5 April 2022) by Diamond (version 0.8.35) with E-values of $\leq 1 \times 10^{-5}$ [84].

**Phylogenetic analyses.** The evolutionary relationships of *T. asperellum* ND-1 and other selected fungi species were evaluated using the proteomes of these fungi. Protein sequence alignment was performed using ClustalW software [85], and the phylogenetic tree was constructed by MEGA version 7.0 [86] with the UPGMA method. In addition to *T. asperellum* ND-1, the proteomes of other selected fungi available on DOE Joint Genome Institute [87] were contained: *T. asperellum* CBS 433.97 (GenBank assembly accession GCA_003025105.1), *T. reesei* QM6a (GCF_000167675.1), *T. virens* Gv-298 (GCF_000170995.1), *T. harzianum* CBS 226.95 (GCA_003025095.1), *T. longibrachiatum* ATCC 18648 (GCA_003025155.1), *T. atroviride* IMI 206040 (GCF_000171015.1), *T. guizhouense* NJAU 4742 (GCA_002022785.1), *T. gamsii* T6085 (GCF_001481775.2), *T. parareesei* CBS 125925 (GCA_001050175.1), *T. citrinoviride* TUCIM 6016 (GCA_00302 5115.1), *A. niger* CBS 513.88 (GCF_000002855.3), *A. oryzae* 3.042 (GCA_000269785.2), *A. terreus* NIH2624 (GCF_000149615.1), *A. glaucus* CBS516.65 (GCF_001890805.1), *A. aculeatus* ATCC16872 (GCF_001890905.1), *A. flavus* NRRL3357 (GCF_00000627 5.2), *G. frondosa* 9006-11 (GCA_001683735.1), *P. decumbens* IBT 11843 (GCA_002072245.1), *Thermothelomyces thermophila* 42464 (GCF_000226095.1), *F. graminearum* PH-1 (GCF_000240135.3), *Aureobasidium pullulans* EXF-150 (GCA_000721785.1). The neighbor joining method with a Poisson model was used for phylogenetic evaluation, and the reliability of branching order was evaluated by 1000 bootstrap replications.

**Comparison analysis of orthologous gene families.** In order to identify the orthologous genes of the four *Trichoderma* species (*T. asperellum* ND-1, *T. reesei* QM6a, *T. atroviride* IMI 206040, *T. virens* Gv-298), we used orthoMCL for the similar pairwise matches to confirm that the groups were orthologous in the *Trichoderma* genomes [65,88]. The genes that were defined as orthologs from clusters of paralogs were subtracted, then the rest of species-specific gene sets of the cluster group expanded because of the most recent common ancestor (MRCA) of the four *Trichoderma* genomes [43].

**Carbohydrate-active enzymes identification and analysis.** For the detection of CA-Zymes, the families of structurally related catalytic (glycosyltransferases (GTs), carbohydrate esterases (CEs), glycoside hydrolases (GHs), auxiliary activities (AAs), polysaccharide lyases (PLs)), and carbohydrate-binding modules (CBMs)) in the four *Trichoderma* species (*T. asperellum* ND-1, *T. reesei* QM6a, *T. atroviride* IMI 206040, and *T. virens* Gv-298) were analyzed exactly based on the CAZymes database (http://www.cazy.org/, accessed on 5 April 2022). A HMMER version 3.3 scan (http://www.hmmer.org/, accessed on 5 April 2022) was performed for annotated CAZyme domain boundaries according to the

dbCAN CAZyme domain HMM database [89,90]. DIAMOND (http://www.dia-mondsearch.org/, accessed on 5 April 2022) and Hotpep were used for fast blast hits in the CAZy database and short conserved motifs in the Peptide Pattern Recognition (PPR) library, respectively. The Hotpep combined with peptide patterns was performed to acquire a stand-alone software for functional prediction and annotation of protein models corresponding to CAZymes.

**Enzyme assays.** β-glucosidase, β-xylosidase, α-L-arabinofuranosidase, and cellobiohydrolase activities were evaluated [91]. Reaction mixture containing 100 μL crude enzyme and 100 μL of 5 mM pNPX, pNPC, pNPG, and pNPAf substrates were incubated in 50 mM sodium acetate buffer (pH 5.0) at 50 °C for 10 min. The reaction was terminated using 100 μL sodium carbonate (1.0 M). A mixture without enzymes was used as the control. An amount of liberated pNP was quantified by determining the absorbance at 405 nm, and one unit was defined as the number of enzymes required to release 1 μmol pNP per min.

Endoxylanase and endoglucanase activities were assayed using the 3,5-dinitrosalicylic acid (DNS) method [92], with 1% (*w/v*) of CMC-Na and BWX as substrate, respectively. The reaction system (150 μL of 1.0% (*w/v*) substrate with 50 μL crude enzyme) was incubated in 50 mM sodium acetate buffer (pH 5.0) for 10 min at 50 °C, and the reaction was stopped by adding 50 μL of 1 M NaOH. A mixture without enzymes was used as the control. After boiling at 100 °C for 5 min, the amount of reducing sugar was assayed at absorbance 540 nm, with one activity unit defined as the enzyme (endoglucanase or endoxylanase) amount that liberated 1 μmol of reducing sugar (equivalent to glucose or xylose) per min from CMC-Na or BWX (equivalent to glucose or xylose) per min under assay conditions. The respective standard curves were obtained with 0.1–0.7 mg/mL glucose and xylose. All enzyme activities were performed in triplicate.

**Supplementary Materials:** The following supporting information can be downloaded at: https://www.mdpi.com/article/10.3390/catal12040437/s1, Figure S1. Percentage distribution of different types of repetitive elements in the genome of *T. asperellum* ND-1.; Figure S2. Functional annotation of the *T. asperellum* ND-1 secretome showing top 20 hits of different category. MF, molecular function; CC, cellular component; BP, biological process; Figure S3. Statistical analysis of CAZymes of *T. asperellum* ND-1 genome. Different colors of the pie chart represent different CAZy classifications, and their areas represent the proportion of genes in the classification; Table S1. Genome features of *T. asperellum* ND-1, *T. asperellum* IC-1 and *T. asperellum* CBS 433.97.; Table S2. Repetitive elements identified in the *T. asperellum* ND-1 genome. Table S3. Glycoside hydrolases (GHs) identified in the genome of *T. atroviride* IMI 206040, *T. virens* Gv-298, *T. reesei* QM6a and *T. asperellum* ND-1.

**Author Contributions:** F.Z. designed the experiments. F.Z., T.H., and J.L. performed the experiments. F.Z. and T.M. analyzed the data. F.Z. and A.B. wrote the original and revised manuscripts. W.J. supervised the present study. All authors have read and agreed to the published version of the manuscript.

**Funding:** This research was funded by National Natural Science Foundation of China, grant number 31570067.

**Data Availability Statement:** The whole genome sequence data reported in this paper has been deposited in the Genome Warehouse [93] at the National Genomics Data Center, Beijing Institute of Genomics (China National Center for Bioinformation), Chinese Academy of Sciences, under accession number GWHAOQT00000000, which is publicly accessible at https://bigd.big.ac.cn/gwh, accessed on 5 April 2022.

**Conflicts of Interest:** The authors declare no competing interest.

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
