# Peer review of "Whole-Genome Sequence and Comparative Analysis of Trichoderma asperellum ND-1 Reveal Its Unique Enzymatic System for Efficient Biomass Degradation"

_catalysts, doi:10.3390/catal12040437_

Round 1

Reviewer 1 Report

The manuscript of a research paper entitled “Whole-genome sequence and comparative analysis of Trichoderma asperellum ND-1 reveal its unique enzymatic system for efficient biomass degradation” by Fengzhen Zheng et al. submitted to Catalysts focuses on analysis of genomic sequence, secretome and comparative analysis of Trichodaerma sp. adeptness of degradation of lignocellulosic biomass.

General remarks:

The subject of the manuscript is in accordance with the aims and scope of Catalysts as it describes an important biological catalyst for green chemistry. The novelty of the work is not exceptionally high as the full genome sequences of several Trichoderma sp. are available. The genetic and transcriptomic information have been analysed to reveal biomass conversion power also on the potential of the organisms to degrade lignocellulose substrates. The detailed secretome analysis gives the value for the manuscript.

The manuscript is generally adequately structured and composed, and generally clearly presented but some additions or clarifications have to be made before the manuscript can be considered for publication. The manuscript is generally written in the academic style English language but English grammar editing is encouraged. The authors should pay attention to the formatting according to the instructions of Catalysts.  

Specific comments:

Title. It is not clear what the “unique enzymatic system” is referred to. It could be clarified in the Abstract.

Introduction (p 1-2). Currently the correct name is “lignin-modifying enzymes” instead of somewhat outdated term “ligninases”. The information regarding rather novel group of enzymes – lytic polysaccharide monooxygenases (LPMOs) – that are vital carbohydrate-active enzymes in lignocellulose (especially cellulose) degradation in Trichoderma sp. is lacking. More information and vital details should be provided on LPMOs to Introduction and in the Results section it should be clarified that enzymes classified as auxiliary activity enzymes are actually LPMOs.

Results and discussion. P 2 and 3 (Table 1), p 14, p 16 – the font sizes have to be amended according to the instructions. The features and quality of the figures have to be unified and more similar to each other. P 5 “A large of extracellular enzymes from T. asperellum have been recognized …” – a word seems to be missing in the sentence. P 7 – The figure legend of Fig. 4 should be corrected and relevant details added.

Materials and methods. P 14 – the salt hydrates are not correctly presented.

Supplementary information has not been submitted for the review and it was not possible to inspect relevant information from Tables S1, S2, S3 and Figures S1, S2, S3.

Author Response

Dear Editors and Reviewers,

Thank you for your carefully consideration and constructive comments for our manuscript with title “Whole-genome sequence and comparative analysis of Trichoderma asperellum ND-1 reveal its unique enzymatic system for efficient biomass degradation”. The manuscript has been revised and significantly improved according to your suggestions.

The response to your comments are given below.

Point 1: Title. It is not clear what the “unique enzymatic system” is referred to. It could be clarified in the Abstract.

Response 1: Thank you. The “unique enzymatic system” is referred to hemicellulases (particularly xylanases) and chitinases. “Abstract” have been revised to clarify the “unique enzymatic system”. Please check. 

Point 2: Introduction (p 1-2). Currently the correct name is “lignin-modifying enzymes” instead of somewhat outdated term “ligninases”. The information regarding rather novel group of enzymes – lytic polysaccharide monooxygenases (LPMOs) – that are vital carbohydrate-active enzymes in lignocellulose (especially cellulose) degradation in Trichoderma sp. is lacking. More information and vital details should be provided on LPMOs to Introduction and in the Results section it should be clarified that enzymes classified as auxiliary activity enzymes are actually LPMOs.

Response 2: Your suggestions are highly appreciated. The “ligninases” has has been replaced with “lignin-modifying enzymes” in the revised Introduction, please see page 1-2. 

LPMOs are the monocopper enzymes widely distributed in nature that catalyses the hydroxylation of glycosidic bonds in most abundant available polysaccharide in nature i.e. cellulose. According to your suggestion, we have added respectively the following contents in the Introduction and the Results section: 

1)“Moreover, the discovery of lytic polysaccharide monooxygenases (LPMOs) has profoundly changed the way in which we view the enzymatic conversion of polysaccharides, particularly recalcitrant materials such as cellulose and chitin (21). LPMOs have been classified in the CAZymesdatabase, within the Auxiliary Activity (AA) families AA9-11 and AA13-16, on account of their sequences (22-24). The most widely investigated LPMO families are AA9 and AA10 (25). LPMOs are currently known to be encoded in genomes across all kingdoms of life, especially fungi, and catalyze cleavage of various substrates (26-29). In addition, LPMOs may be subjected to various post-translational modifications depending on their origin with effects on protein function and stability (30).” Please see page 2 with highlight in the revised Introduction.

2)“LPMOs are the monocopper enzymes widely distributed in nature that catalyses the hydroxylation of glycosidic bonds in most abundant available polysaccharide in nature i.e. cellulose (57,58). Secretomic analysis revealed that asperellumND-1 encode two predicted LPMOs from AA9 and AA11, respectively. Moreover, AA9 family could have important roles as copper-dependent LPMOs, cleaving oxidatively biomass cellulose (38). Additionally, this work contributes to the broader mapping of enzyme activity in Auxiliary Activity family (particularly AA9, AA11 and AA14) and provides new biocatalysts for potential applications in biomass modification. “ Please see page 5 with highlight in the revised Results section.

Point 3: Results and discussion. P 2 and 3 (Table 1), p 14, p 16 – the font sizes have to be amended according to the instructions. The features and quality of the figures have to be unified and more similar to each other. P 5 “A large of extracellular enzymes from T. asperellum have been recognized …” – a word seems to be missing in the sentence. P 7 – The figure legend of Fig. 4 should be corrected and relevant details added.

Response 3: Thank you for the detailed suggestions. The font sizes in P 2 and 3 (Table 1), p 14, p 16 have been modified, please check. 

The features and quality of the figures have been revised, please check.

“A large of extracellular enzymes from T. asperellum have been recognized …” has been replaced with “A large of extracellular enzymes secreted from T. asperellum have been recognized …”, Please see page 5 with highlight. 

The figure legend of Fig. 4 has be corrected and added the following content :

“The tree of selected genome sequences was constructed by using the neighbor-joining (NJ) method with Poisson model as implemented in Mega software 7.0. T. asperellum ND-1 was labeled with diamonds.” Please see the figure legend of Fig. 4 with highlight. 

Point 4: Materials and methods. P 14 – the salt hydrates are not correctly presented.

Response 4: Thank you for the constructive comments. The salt hydrates have been revised, Please see page 14 with highlight. 

Point 5: Supplementary information has not been submitted for the review and it was not possible to inspect relevant information from Tables S1, S2, S3 and Figures S1, S2, S3.

Response 5: Thank you. The Supplementary information has been submitted, please check.  

We believe that all the recommendation and suggestions improved the quality of the manuscript.

Reviewer 2 Report

Review (minor revision):

The article “Whole-genome sequence and comparative analysis of Trichoderma asperellum ND-1 reveal its unique enzymatic system for efficient biomass degradation” by Zheng et al., discusses the genome of the Trichoderma asperellum ND-1 strain. More accurately, the authors present a high-accuracy genome sequence of the mentioned strain.

Introduction:

The introduction part describes the topics of the manuscript well. Moderate improvements in English are necessary.

Materials and methods:

How were the crude enzymes procured?

Conclusion:

Some minor English improvements are needed. Other than that, the conclusion part is written well.

Author Response

Dear Editors and Reviewers,

Thank you for your carefully consideration and constructive comments for our manuscript with title “Whole-genome sequence and comparative analysis of Trichoderma asperellum ND-1 reveal its unique enzymatic system for efficient biomass degradation”. The manuscript has been revised and significantly improved according to your suggestions.

The response to your comments are given below.

Point 1: Introduction: 

The introduction part describes the topics of the manuscript well. Moderate improvements in English are necessary.

Response 1: Thank you. The manuscript has been revised by an English language expert and all the grammatical mistakes have been removed.

Point 2: Materials and methods:

How were the crude enzymes procured?

Response 2: Your suggestions are highly appreciated. According to your suggestion, we have added the following contents in the Materials and methods: 

“for 6 days. Culture samples were taken every day and centrifuged at 12,000×g for 10 min to collect the supernatant. Then the supernatant containing the crude enzymes was used directly for enzyme assays.” Please see page 14 with highlight.

Point 3: Conclusion:

Some minor English improvements are needed. Other than that, the conclusion part is written well.

Response 3: Thank you. The manuscript has been revised by an English language expert and all the grammatical mistakes have been removed.

We believe that all the recommendation and suggestions improved the quality of the manuscript.

Round 2

Reviewer 1 Report

Review to the revised manuscript entitled “Whole-genome sequence and comparative analysis of Trichoderma asperellum ND-1 reveal its unique enzymatic system for efficient biomass degradation” by Fengzhen Zheng et al. submitted to Catalysts.

I sincerely thank the authors for their effort. The manuscript has been considerably improved during the revision and the Supplementary figures and tables are made available.

The main issues and comments that were pointed out have been addressed by the authors. The figures still seem rather different from each other regarding the preparation and quality but they are readable. The manuscript is sufficiently updated to warrant publication.